# The Association of MEG3 lncRNA with Nuclear Speckles in Living Cells

**DOI:** 10.3390/cells11121942

**Published:** 2022-06-16

**Authors:** Sarah E. Hasenson, Ella Alkalay, Mohammad K. Atrash, Alon Boocholez, Julianna Gershbaum, Hodaya Hochberg-Laufer, Yaron Shav-Tal

**Affiliations:** The Mina & Everard Goodman Faculty of Life Sciences and The Institute of Nanotechnology and Advanced Materials, Bar-Ilan University, Ramat Gan 5290002, Israel; hasenss@biu.ac.il (S.E.H.); ella.alkalay@live.biu.ac.il (E.A.); atrashm1@biu.ac.il (M.K.A.); boochoa@biu.ac.il (A.B.); jgershbaum@gmail.com (J.G.); hodayahoc@gmail.com (H.H.-L.)

**Keywords:** lncRNA, MEG3, nuclear bodies, nuclear speckles, nucleus, splicing, transcription, live-cell imaging

## Abstract

Nuclear speckles are nuclear bodies containing RNA-binding proteins as well as RNAs including long non-coding RNAs (lncRNAs). Maternally expressed gene 3 (MEG3) is a nuclear retained lncRNA found to associate with nuclear speckles. To understand the association dynamics of MEG3 lncRNA with nuclear speckles in living cells, we generated a fluorescently tagged MEG3 transcript that could be detected in real time. Under regular conditions, transient association of MEG3 with nuclear speckles was observed, including a nucleoplasmic fraction. Transcription or splicing inactivation conditions, known to affect nuclear speckle structure, showed prominent and increased association of MEG3 lncRNA with the nuclear speckles, specifically forming a ring-like structure around the nuclear speckles. This contrasted with metastasis-associated lung adenocarcinoma (MALAT1) lncRNA that is normally highly associated with nuclear speckles, which was released and dispersed in the nucleoplasm. Under normal conditions, MEG3 dynamically associated with the periphery of the nuclear speckles, but under transcription or splicing inhibition, MEG3 could also enter the center of the nuclear speckle. Altogether, using live-cell imaging approaches, we find that MEG3 lncRNA is a transient resident of nuclear speckles and that its association with this nuclear body is modulated by the levels of transcription and splicing activities in the cell.

## 1. Introduction

The nucleus of higher eukaryotes contains various membraneless sub-nuclear compartments termed nuclear bodies [1,2]. One such body is the nuclear speckle, which is host to many different types of RNA-processing factors as well as lncRNAs [3,4,5]. While most nuclear speckle components are splicing factors, many factors involved in the gene expression pathway can be found within. These include factors involved in transcription, RNA modifications and mRNA export regulation. Nuclear speckles were first identified by electron microcopy as 25–50 non-randomly distributed irregularly shaped nuclear structures of varying sizes and were named inter-chromatin granule clusters (IGCs). The components of IGCs were later discovered to be factors involved mostly in pre-mRNA splicing [4,6]. Different sub-regions are identified within the nuclear speckles, as determined by super-resolution microscopy, and different components are detected in them [7]. For instance, proteins such as Serine/Arginine Repetitive Matrix 2 (SRRM2) and SON, both RNA-binding proteins and known markers for nuclear speckles, can be found in the core region of the structure, while the splicing factor small nuclear ribonucleoprotein polypeptide B2 (U2B), non-coding RNAs such as MALAT1 and small nuclear RNAs (snRNAs) localize at the periphery of the nuclear speckles.

Gene expression can be regulated by nuclear speckles but their exact roles are unclear [4]. It is suggested that nuclear speckles can function as storage and recycling sites for splicing factors [8,9]. Live-cell imaging studies have demonstrated splicing factors leaving the nuclear speckles supposedly to be recruited to transcription sites while fluorescence recovery after photobleaching (FRAP) analysis has shown that factors are constantly moving in and out of these structures [10,11]. Moreover, both transcription inhibition and splicing inhibition lead to the accumulation of splicing factors in the nuclear speckles, which then grow in size and take on a rounder shape instead of the typical irregular structure, suggesting that under conditions with no splicing, the splicing factors only enter the nuclear speckles but do not leave [12,13]. We recently proposed that nuclear speckles can take part in buffering the levels of splicing factors in the nucleoplasm, thereby regulating the amount of splicing factors that are available for active splicing [14].

Nuclear speckles are also suggested to function as ‘gene expression hubs’ since they were shown to have an enhancing effect on transcription and RNA-processing activities for certain genes [15,16]. This may be influenced by the degree of proximity between the gene and the nuclear speckle, and by the recycling of the necessary factors between the gene and the nuclear speckles [11,17,18,19,20,21]. Interestingly, even though nuclear speckles may augment transcription of certain genes, they typically do not contain active genes nor DNA within them. Therefore, nuclear speckles might be facilitating an enhancement of gene expression by forming a local high concentration of different factors.

Staining for poly(A) tails of transcripts has shown that nuclear speckles contain a substantial population of poly(A)+ RNAs [22,23,24]. When transcription was inhibited with α-amanitin, no significant changes in the poly(A)+ RNA population were observed, indicating that the poly(A)+ RNAs were not messenger RNAs (mRNAs) [23,25,26,27]. The lncRNA MALAT1 was the first poly(A)+ RNA identified as a key resident of nuclear speckles, whereas the related lncRNA Nuclear Enriched Abundant Transcript 1 (NEAT1) was found to localize in associated structures termed paraspeckles [28,29,30,31]. Most lncRNAs are nuclear retained and can also reside in sub-nuclear compartments [32]. MALAT1 is nuclear retained and its association with the nuclear speckles is transcription dependent and its primary function is alternative splicing regulation [33,34,35]. Many lncRNAs are transcribed by RNA polymerase II and contain a 5’-cap and a 3’ poly(A) tail. Generally, lncRNAs are expressed at lower levels than protein-coding mRNAs, and they are more tissue or cell type specific [32,36,37].

Extensive studies have revealed functions for some lncRNAs, but the vast majority remain uncharacterized even at the most basic levels, such as subcellular localization and absolute abundance. In many cases, lncRNAs localize at their site of action, which therefore can provide insights into their biological function. Single-molecule fluorescence in situ hybridization (smFISH) was used to identify the expression patterns of lncRNAs including MEG3 [37,38]. This gene is located on chromosome 14 in human cells and its expression is regulated by differentially methylated promoters from the *DLK1-MEG3* imprinting locus on the same chromosome [39,40,41]. MEG3 is expressed at high levels in the brain, placenta and endocrine glands [42,43,44]. MEG3 is nuclear retained due to its nuclear retention element (NRE), a 365 nucleotide (nt) element, which spans exon 3, 4 and 6 [45]. Its removal resulted in the partial cytoplasmic localization of MEG3. MEG3 is involved in different pathways and is expressed in many diseases as well as having tumor suppressor activity by selective activation of p53 response [46]. The underlying causes for loss of MEG3 expression in tumors are many, such as gene deletion, promoter hypermethylation and hypermethylation of the intergenic regions [47]. MEG3 was identified to partially associate with the nuclear speckles [37] and since lncRNAs often localize to their site of action, this localization could shed light on the connections of MEG3 to the gene expression pathway.

In this study, we examined the dynamics of MEG3 lncRNA in the nucleus of living cells. MEG3 is expressed in primary cells, while in cancer cell lines it is usually expressed at low levels. Therefore, we generated an inducible MEG3 gene that contains elements for tagging RNA in living cells, to be expressed in a cancer cell line. We find that MEG3 is a lncRNA that can transiently associate with nuclear speckles under regular conditions, but when transcription or splicing inhibition are inhibited, the association with the periphery of nuclear speckles becomes predominant, forming a distinct ring-like region surrounding the nuclear speckle.

## 2. Materials and Methods

### 2.1. Cell Culture

U2OS Tetracycline-On (Tet-On) human osteosarcoma cells were maintained in low-glucose Dulbecco’s modified Eagle’s medium (DMEM) (Biological Industries, Beit-Haemek, Israel) containing 10% FBS (HyClone Laboratories, Logan, UT, USA). Stable expression of MEG3-MS2 + Yellow Fluorescent Protein (YFP)-MS2 coat protein (CP) [48] was obtained using PolyJet^TM^ (SignaGen Laboratories, Frederick, MD, USA) transfection and selection with puromycin (1 μg/mL, Invivogen, San Diego, CA, USA). Cells were grown at 37 °C and 5% CO_2_. HFF-1 (human foreskin fibroblast; provided by Ron Goldstein, BIU, Ramat Gan, Israel) cells were maintained in high-glucose DMEM (Biological Industries) containing 10% FBS. HepG2 (human liver cancer; provided by Uri Nir, BIU, Ramat Gan, Israel) cells were maintained in MEM-EAGLE Earle’s Salt Base medium (Biological Industries) containing 10% FBS. U2OS Tet-On cells were purchased from Clonetech (Mountain View, CA, USA). A STR profile was recently conducted on them. Mycoplasma tests were carried out periodically on all cell lines. MEG3-MS2 transcription was induced by the addition of doxycycline (dox) (15 µg/mL) to the medium for 48 h.

For transcription inhibition, cells were grown on coverslips and incubated at 37 °C for 2 h with either actinomycin D (ActD) (5 µg/mL, Sigma, Rehovot, Israel) or 5,6-dichloro-1-β-D-ribofuranosylbenzimidazole (DRB) (50 µg/mL, Sigma) or 6 h with α-amanitin (30 µg/mL, Sigma) [49] before fixation for 20 min in 4% paraformaldehyde (PFA). To release from the DRB treatment, cells were washed with fresh medium for 30 min at 37 °C before fixation. For splicing inhibition, cells were grown on coverslips and incubated at 37 °C for 6 h with Pladienolide B (PLB) (0.5 µM, Santa Cruz Biotechnology, Dallas, TX, USA) [50] before fixation for 20 min in 4% PFA. To release from the PLB treatment, cells were washed with fresh medium for 30 min at 37 °C before fixation.

### 2.2. Plasmids and Transfections

Cloning of MEG3 (NR_033358) under Tet-On control: A plasmid containing MEG3 isoform 4 was synthesized (Rhenium, Modi’in-Maccabim-Re’ut, Israel) [51] and cloned into the *p*-tetracycline response element (pTRE) plasmid containing 24x MS2 repeats, by polymerase chain reaction (PCR) amplifying the MEG3 sequence with primers matching the beginning and the end sequence with the addition of a BsrGI restriction site. The pTRE-24x-MS2 plasmid and the MEG3 PCR product were ligated following BsrGI restriction. The pTRE MEG3-MS2 plasmid was stably co-transfected with a YFP-MS2-A1 coat protein plasmid [48] into U2OS Tet-On cells together with a plasmid for puromycin antibiotic selection. Different MEG3-MS2 cell clones were generated and used in the study. The pTRE-MEG3 plasmid without MS2 repeats was created with: forward primer 5’-ACAAAGACGCGTTGTACAAGCCCCTAGC-3’ matching the beginning of the exon 1 sequence and reverse primer 5’-GGGGGGGATCGATTTTTTTTTGTTAAGAC-3’ matching the end of exon 8. The forward primer contained the restriction site for MluI and the reverse primer contained the restriction site of ClaI for insertion into the pTRE plasmid. Truncated MEG3 containing exons 1–3 (MEG3 Ex1–3) was created with: forward primer 5’-AAGCTTTGTACAAGCCCCTAGCGC-3’ matching the beginning of the sequence and reverse primer 5’-TAAGGTGTACAGCTGATGCAAGGA-3’ matching the end of exon 3. Both primers contained the BsrGI restriction site for insertion into the pTRE-24x-MS2 plasmid. Truncated MEG3 containing part of the NRE found on exon 3 (MEG3 Ex3 NRE) was created with: forward primer 5’-ACGGTACTGTACAGGAGGTGATCAGCAA-3’ matching the beginning of the NRE and reverse primer 5’-TAAGGTGTACAGCTGATGCAAGGA-3’ matching the end of exon 3. Both primers contained the BsrGI restriction site for insertion into the pTRE-24x-MS2 plasmid. pTRE-MEG3, pTRE-MEG3 Ex3 NRE and pTRE-MEG3 Ex1-3-MS2 were transfected using Lipofectamine 2000 (Thermo Fisher Scientific, Waltham, MA, USA) or Polyjet (SignaGen Laboratories) according to the manufacturer’s instructions.

### 2.3. Immunofluorescence

Cells were grown on 18 mm coverslips, washed with phosphate-buffered saline (PBS) and fixed for 20 min in 4% PFA. U2OS Tet-On cells were then permeabilized in 0.5% Triton X-100 for 2 min, after which they were blocked in bovine serum albumin (BSA). HFF-1 and HepG2 were permeabilized twice in 0.1% Triton X-100 for 5 min. Cells were immunostained for 1 hr with a primary antibody, and after subsequent washes the cells were incubated for 45 min with secondary fluorescent antibodies. Primary antibodies: Mouse anti-Serine and Arginine Rich Splicing Factor 2 (SRSF2) (which marks SRRM2 [52]), rabbit anti-SON (Sigma), rabbit anti-Heterogeneous Nuclear Ribonucleoprotein K (hnRNPK) (Abcam), rabbit anti-SRRM2 (Abcam) and rabbit anti-Serine and Arginine Rich Splicing Factor 7 (SRSF7) (Santa Cruz). Secondary antibodies: Alexa647-labeled goat anti-mouse IgG, Alexa594-labeled goat anti-mouse IgG, Alexa647-labeled donkey anti-rabbit IgG (Life Technologies, Thermo Fisher Scientific), dyLight488-labeled goat anti-rabbit IgG and Alexa488-labeled goat anti-mouse IgG (Abcam). The cells were mounted in mounting medium.

### 2.4. Fluorescence Microscopy

Wide-field fluorescence images were obtained using the CellSens system based on an Olympus IX81 fully motorized inverted microscope (60x Planpon O objective 1.42 [NA]) fitted with an Orca-Flash4.0 #2 camera (Hamamatsu, Bridgewater, NJ, USA), rapid wavelength switching, and driven by the CellSens v3.2 software (Olympus, Tokyo, Japan); or the CellR system based on an Olympus IX81 fully motorized inverted microscope (60x PlanApo objective 1.42 [NA] Olympus) fitted with an Orca-AG CCD camera (Hamamatsu), rapid wavelength switching, and driven by the CellR v1.2 software (Olympus). For time-lapse imaging, cells were plated on glass-bottomed tissue culture plates (MatTek, Ashland, MA, USA) in DMEM medium containing 10% fetal calf serum at 37 °C. The microscope is equipped with an on-scope incubator which includes temperature and CO_2_ control (Life Imaging Services, Reinach, Switzerland). For long-term imaging, several cell positions were chosen and recorded by a motorized stage (Scan IM; Märzhäuser, Wetzlar-Steindorf, Germany). Cells were typically imaged in three dimensions (3D) (11 Z planes per time point) every 5 or 10 min. For quantifications of MEG3 intensity in association with the nuclear speckle, the cells were imaged in 31 Z planes per time point. Rapid live-cell imaging was performed by imaging every 300 msec for 60 s in one Z plane. Live-cell imaging and 3D stacks were deconvolved using Huygens Essential v18.10 software (Scientific Volume Imaging, Hilversum, Netherlands) and analyzed by Imaris v9.8.2 software (Oxford Instruments, Oxfordshire, UK). The perimeter of the nuclear speckle area analyzed with the Imaris extended outward with a threshold value of 0.25 or 0.25 µm from the surface of the nuclear speckles. Colocalization analysis of two channels was performed using an ImageJ Macro (Shav-Tal lab, Ramat Gan, Israel).

Super-resolution fluorescence and confocal images were obtained using the Leica SP8 Stimulated Emission Depletion (STED) confocal microscope (63x HC PL APO objective 1.40 [NA]) with the LASX v3.5.5 software (Leica, Wetzlar, Germany). For STED imaging, two continuous-wave lasers at 592 nm and 660 nm were used, which allow the use of fluorophores up to 595 nm in the super-resolution mode. The white light laser (WLL) function was used for the confocal imaging.

FRAP experiments were performed using a Leica Laser scanner based on Leica Total Internal Reflection Fluorescence (TIRF) microscope (63x HC PL APO objective 1.40 [NA]) with the LASX v3.7.5 software (Leica,), as previously described [14]. Before and after the bleaching, cells were imaged in the YFP channel for the detection of YFP-MS2 (MEG3) and in the Cyan Fluorescent Protein (CFP) channel for the detection of Cerulean Fluorescent Protein coupled to Serine and Arginine Rich Splicing Factor 3 (Cer-SRSF3) (nuclear speckle factor). MEG3 signal at a specific nuclear speckle was photobleached using the 553 nm laser. Six pre-bleach images were acquired. Post-bleach images were acquired in a sequence of 3 time frequencies: 15 images every 3 s, 15 images every 6 s, and 26 images every 30 s. For each time point, the background taken from a region of interest (ROI) outside of the cell was subtracted from all other measurements. *T*_(*t*)_ and *I*_(*t*)_ were measured for each time point as the average intensity of the nucleus and the average intensity in the bleached ROI, respectively. The average of the pre-bleach images used as the initial conditions are referred to as *T_i_* = nuclear intensity and *I_i_* = intensity in the ROI before bleaching. *Ic*_(*t*)_ is the corrected intensity of the bleached ROI at time *t*:$$Ic{\;}_{\left(t\right)\;}=\frac{I_{\left(t\right)}T_{i}}{I_{i}T_{\left(t\right)}}$$

The diffusion of the YFP-MS2 protein during the FRAP analysis was disregarded since the diffusion rate of free nucleoplasmic YFP-MS2 is very rapid, while the bound YFP-MS2 is associated with high affinity to the RNA and does not detach or diffuse [53,54].

### 2.5. RNA FISH

Cells were seeded on 18 mm coverslips and fixed for 20 min in 4% PFA, then permeabilized twice in 0.1% Triton X-100 for 5 min. Coverslips were then washed twice with 10% formamide for probes purchased from Stellaris or 15% formamide for FLAP [55] probes diluted in 4x Saline-sodium citrate (SSC). For smFISH on endogenous transcripts, fluorescent-labeled DNA probes that target the MEG3 sequence (570 nm, ~10 ng probe, Stellaris; or 670 nm, ~5 ng probe, FLAP), MALAT1 sequence (570 nm, ~10 ng probe, Stellaris) or NEAT1 sequence (570 nm, ~10 ng probe, Stellaris) were hybridized overnight at 37 °C in a dark chamber in 10% formamide or 15% formamide, respectively. The next day, cells were washed twice with 10% formamide for Stellaris probes or 15% formamide for FLAP probes diluted in 4x SSC for 30 min at 37 °C and then washed with 1x PBS. The coverslips were mounted with mounting medium.

For MS2 RNA FISH, cells were left overnight in 70% ethanol at 4 °C and then fluorescently labeled probes that target the MS2 repeats (670 or 570, ~10 ng probe per coverslip) were hybridized overnight at 37 °C in a dark chamber in 40% formamide. The next day, cells were washed twice with 40% formamide diluted in 4x SSC for 15 min followed by two additional washes with 1x PBS for one hr each. Coverslips were mounted with mounting medium.

### 2.6. SiRNA Knockdowns

Cells were transfected with small interfering RNAs (siRNA) to knock down SON (Thermo Fisher Scientific), hnRNPK (IDT, Jerusalem, Israel), SRSF7 (Invitrogen, Carlsbad, CA, USA), SRRM2 (IDT) or a negative scrambled control (IDT), using Lipofectamine 2000.

### 2.7. FACS Cell Cycle Analysis

A total of 10^6^ wild-type or MEG3 U2OS Tet-On cells were seeded in a 10 cm plate. Dox (15 µg/mL) was added to the medium for 48 h. Trypsin was used to detach cells from plates. Cells were washed with PBS and then fixed in 4% PFA for 15 min. Cells were washed and resuspended in 4’,6-diamidino-2-phenylindole (DAPI) (1 mg/mL) in PBS and 10% Triton X-100 at room temperature for 8 min. Cells were washed and the pellet was resuspended with 1 mL PBS and analyzed in a Flow Cytometer LSRFortessa (BD Biosciences, NJ, USA). Analysis of the data output was performed with the FlowJo v10.8 software (Ashland, OR, USA) using the Watson model to calculate the percent of cells in each stage of the cell cycle.

### 2.8. Statistical Analysis

All statistical analyses were performed using R statistical software v4.1 (Oxford Protein Informatics Group, Oxford, United Kingdom) (R Core Team (2021). R: A language and environment for statistical computing. R Foundation for Statistical Computing, Vienna, Austria. https://www.R-project.org/). Three-parameter asymptotic regression was used to analyze FRAP experiments. Each replicate (3) was fit to a curve defined by the equation *Y = a* − (*a* − *b*) *e^−cX^*, where *X* is time and *Y* is the relative intensity, a (*plateau*) is the maximum attainable intensity, b (*init*) is the initial *Y* value (at time = 0) and c (*m*) is proportional to the relative rate of increase for intensity when time increases. The regression fitting was performed using the *drm* function from the drc R package [56] and DRC. asymReg self-starting function from the aomisc R package [57] and The broken bridge between biologists and statisticians: a blog and R package, Statforbiology, IT, web: https://www.statforbiology.com). Each parameter (*init*, *m* and *plateau*) was then compared between all treatments with a one-way nested ANOVA, followed by pairwise comparisons of mean values between treatments. Finally, *p* values were adjusted for multiple comparisons with the Benjamini–Hochberg (FDR) procedure. The function *lmer* from lmerTest R package [58], and *emmeans* from emmeans R package (Russell V. Lenth (2021). emmeans: Estimated Marginal Means, aka Least-Squares Means. R package version 1.7.1-1. https://CRAN.R-project.org/package=emmeans) were used.

## 3. Results

### 3.1. Detection of MEG3 lncRNA in Living Cells

To facilitate the detection of MEG3 lncRNA in living cells, we generated an osteosarcoma U2OS cell line in which an inducible MEG3 gene was stably expressed. Most cancerous cell lines express MEG3 lncRNA at very low levels and so osteosarcoma cells do not have background levels of MEG3 [59,60,61]. The gene sequence contains a series of 24x MS2 sequence repeats at the 3′-end, used to facilitate tagging of RNA in living cells [62,63]. The expression of MEG3-MS2 was induced under the Tet-On promotor system following the addition of dox to the cell medium (Figure 1A). The MEG3-MS2 gene was co-transfected with the YFP-MS2 coat protein (YFP-MS2-CP) that specifically binds to the repeated stem–loop structures in the 3′-end of the transcript formed by the 24x MS2 repeats, thus allowing for detection of the RNA in living cells (Figure 1B). Single transcripts (lncRNPs) could be detected in the nucleoplasm of the dox-induced U2OS cells and were excluded from nucleoli. The active site of transcription of the MEG3-MS2 gene was also observed. Expression of MEG3-MS2 did not affect the cell cycle of U2OS cells (Appendix A). The distribution patterns of MEG3-MS2 lncRNA in U2OS cells were predominantly nuclear and similar to those of the endogenous MEG3 transcript as observed in HepG2 cells, a cancerous cell line which does express MEG3 [64]. Endogenous MEG3 lncRNA was detected using RNA FISH probes to MEG3 (Appendix A).

No cytoplasmic MEG3 signal was detected in the U2OS cells expressing MEG3-MS2, demonstrating that MEG3 is a nuclear retained lncRNA, as previously documented in normal fibroblasts such as HFF-1 (human foreskin fibroblast), S27 (Human Foreskin Fibroblast), and W138 (Human Lung Fibroblast) cells as well as a cancerous cell line such as hLF (Human Lung Fibroblast Epidermoid Carcinoma) [37,42]. A live-cell movie of MEG3-MS2 tagged with YFP-MS2-CP post-dox addition showed the gradual increase in MEG3 transcription and the distribution of the lncRNPs in the nucleoplasm over a 3-h period (Figure 1B,C, Appendix A). Active transcription was observed about 15 min after dox induction and the lncRNPs began to spread in the nucleoplasm at around 1.5 h and onwards. The transcripts were not evenly distributed throughout the nucleoplasm. We therefore examined whether MEG3-MS2 might be associated with nuclear speckles, as previously suggested [37], like the two other nuclear retained lncRNAs, MALAT1 and NEAT1, which are known to be in association with these nuclear bodies [28]. It was previously shown that lncRNAs and proteins localize within different regions of the nuclear speckle structure. Some proteins are found at the center of the nuclear speckle while MALAT1 lncRNA is found at the periphery [7]. Some of the MEG3 lncRNA signal was in close proximity to the SRRM2 protein, a core nuclear speckle protein (Figure 1D–F). Altogether, there appears to be a partial level of association between the nuclear speckles and the MEG3 signal.

### 3.2. MEG3 Localizes to Nuclear Speckles upon Transcription Inhibition

RNA FISH to MALAT1 lncRNA shows a clear overlap in the position of the lncRNAs and nuclear speckles (Figure 2A), as expected [28]. It was previously shown that MALAT1 and MEG3 colocalize to a high extent in HFF-1 cells [37], and therefore the association of MEG3 with nuclear speckles is expected, although the MEG3 signal appears to be more dispersed compared to the MALAT1 signal, which is more concentrated within the nuclear speckles. To further examine the association of MEG3 with nuclear speckles, we used transcription inhibition conditions. The latter treatment causes nuclear speckles to retain an excess of splicing factors that are no longer needed in the nucleoplasm, leading to the rounding up and enlargement of the nuclear speckles, compared to regular conditions [11,65]. Transcription inhibition was induced by treating cells with DRB that inhibits transcription elongation [66]. While MALAT1 normally resides in the nuclear speckles, transcription inhibition by DRB leads to its dispersal in the nucleoplasm, whereas nuclear speckles remain intact [33] (Figure 2A). Similarly, NEAT1 lncRNA, known to localize to paraspeckles under regular conditions, also disperses after treatment with DRB [67] (Figure 2B). Interestingly, for MEG3 lncRNA, the opposite was observed. Transcription inhibition with DRB led to close localization of MEG3 lncRNAs to the nuclear speckles and a ring-like pattern appeared at the nuclear speckle periphery (Figure 2C). The rounding up of the nuclear speckles could be observed in the DRB-treated cells, as expected. The redistribution of MEG3 in DRB-treated cells was then followed using live-cell imaging after 15 min of treatment. The recruitment of MEG3 to the nuclear speckles began after 35 min of DRB treatment (20 min into the movie) and reached full development by 1 h (Figure 2D and Appendix A).

Different types of transcription inhibitors affect distinct steps of the transcription process. We also tested actinomycin D (ActD) that inhibits the access to DNA for all mammalian RNA polymerases, and α-amanitin that specifically but slowly inhibits RNA polymerase II. Both ActD and α-amanitin had similar effects on MEG3 distribution compared to DRB and localization was not affected by the MS2 repeats (Figure 3A and Appendix A).

The association of protein factors with nuclear speckles is highly dynamic, whereby they are continuously moving in and out of these structures [11,68]. To test whether the MEG3 nuclear speckle localization was reversible, cells were treated with DRB (2 h), and then returned to regular medium for 30 min before fixation (Figure 3B). MEG3 localization to the nuclear speckles caused by transcription inhibition, dispersed after DRB removal. Still, the regular fraction of MEG3 associated with nuclear speckles was observed after release, but to a lesser extent. Quantifications of the MEG3 intensity at the nuclear speckle periphery showed a significant increase both in DRB and in ActD-treated cells compared to untreated or DRB-released cells (Figure 3C,D). Live-cell imaging of cells treated with DRB for 2 h and then released with medium, showed that the reversal of nuclear speckle localization was quite rapid and started 15 min after DRB removal (Figure 3E, Appendix A). The release from DRB was apparent by the return to the typical irregular shape of nuclear speckles.

We verified that the relocalization of MEG3 lncRNA to nuclear speckles also occurs for the endogenous transcript in HepG2 and HFF-1 cells (Figure 4A). These cells were treated with DRB, and endogenous MEG3 transcripts were detected by RNA FISH. In both cell lines there was some MEG3 association with nuclear speckles under regular growth conditions, as seen in U2OS cells. Increased localization to the nuclear speckle periphery was seen upon DRB treatment (Figure 4B).

### 3.3. MEG3 Localizes to Nuclear Speckles during Splicing Inhibition

Nuclear speckles contain a large variety of splicing factors. The phenomenon of rounding up of these structures during transcription inhibition is attributed to the return of splicing factors from the nucleoplasm to the nuclear speckles for recycling without the need to return to genes, since there are no active genes producing pre-mRNAs during transcription inhibition [11,65,69]. Splicing inhibition on its own has the same effect on nuclear speckle structure [70]. We therefore wished to examine whether splicing inhibition per se, without transcription inhibition, can affect the distribution of MEG3 lncRNA. Pladienolide B (PLB) is a compound that inhibits splicing [50] and indeed when the cells were treated with PLB, MEG3 transcripts formed the circular formations surrounding the nuclear speckles, resembling those observed during transcription inhibition. PLB treatment was also reversible but appeared to have somewhat slower dynamics than the release from DRB (Figure 5A,B).

### 3.4. Part of Exon 3 Is Important for MEG3 Localization to Nuclear Speckles

The human *MEG3* gene has eight exons. To examine which sequences in the lncRNA are connected to the localization to nuclear speckles, a shorter clone of MEG3-MS2 was created. We removed roughly half of the MEG3 transcript resulting in a sequence containing exons 1–3 (exon 3 is long compared to the other exons). The transcript was expressed in U2OS cells that were then treated with DRB. Transcripts containing exons 1–3 still localized to the nuclear speckles under DRB conditions, indicating that the last five exons were not required for this localization (Figure 6A,B, Appendix A). Exon 3 contains part of the previously identified nuclear retention element (NRE) of MEG3 [45]. This part of the NRE in exon 3 (138 nts) was cloned and expressed. Nuclear retention was maintained as well as nuclear speckle localization after treatment with DRB (Figure 6C).

As demonstrated above, MEG3 repositioning in relation to the nuclear speckles can take place due to changes in the nuclear speckle structure occurring during transcription and splicing inhibition. Next, we knocked down different RNA-binding factors by use of siRNA, to see if this would affect MEG3 nuclear speckle localization. The core proteins SON and SRRM2 appear to be essential for proper nuclear speckle formation, since their depletion can lead to different degrees of nuclear speckle disassembly or restructuring [52]. When SON or SRRM2 were knocked-down and cells were transcriptionally inhibited, MEG3 still appeared to localize to the nuclear speckles. This would suggest that neither SON nor SRRM2 are independently responsible for the increased speckle localization during transcription/splicing inhibition. Knockdown of the SRSF7 splicing factor or of hnRNPK which is known to bind lncRNAs [71] did not have any impact on MEG3 localization either (Appendix A).

### 3.5. MEG3 Dynamics at the Nuclear Speckle during Transcription Inhibition

We hypothesized that MEG3 once located at the nuclear speckles will remain there as long as the cell is under transcription inhibition conditions. This could be examined by fluorescence recovery after photobleaching (FRAP). Measuring the dynamics of a lncRNA at the nuclear speckle under transcription inhibition has not been examined so far since other lncRNAs such as MALAT1 and NEAT1 are released from the nuclear speckle under transcription inhibition conditions [30,33]. FRAP analysis was performed on cells where MEG3 was tagged with YFP-MS2-CP and co-expressed with Cer-SRSF3 as a nuclear speckle marker. MEG3 signal was photobleached and the return of the YFP signal was measured at the nuclear speckles in untreated and DRB-treated cells. MEG3 signal at the nuclear speckles in untreated cells showed ~90% recovery. In treated cells, the recovery was slower and gradual, indicating increased association of MEG3 with the nuclear speckles (Figure 7A,B and Appendix A). Still, the signal did finally recover indicating that there is a certain degree of MEG3 exchange from the nuclear speckle over time. Next, we further examined this interchange of MEG3 transcripts between the nuclear speckle and the nucleus by tracking of the transcripts. MEG3 movement in relation to the nuclear speckle labeled with Cer-SRSF3 was followed and characterized using rapid live-cell imaging. In transcriptionally inhibited cells, MEG3 movement was confined to the nuclear speckles, whereas the RNA had significantly longer displacement lengths in untreated cells. The limited movements after DRB treatment at the nuclear speckles showed that the transcripts moved back and forth at the nuclear speckle periphery but did not tend to leave its vicinity (Figure 7C,D).

Super-resolution microscopy has shown that nuclear speckles are layered structures and that the SRRM2 and SON proteins are most often found at the core of the nuclear body [7]. To examine the precise position of MEG3 lncRNA at the nuclear speckle and the changes that occur after transcription or splicing inhibition, we stained the core nuclear speckle factors SON and SRRM2, after which the cells were imaged using STED super-resolution microscopy. Before transcription inhibition, MEG3 lncRNA was loosely associated with the periphery of the nuclear speckle core, the latter harboring the SRRM2 and SON proteins. After treatment with DRB or ActD, a prominent MEG3 signal was seen around the nuclear speckles. Interestingly, some MEG3 signal was also detected within the inner part of the nuclear speckle (Figure 8A,B). Still, the RNA and protein signals remained spatially separate even in the core of the nuclear speckle. Confocal imaging of HepG2 cells showed the endogenous MEG3 behaving in a similar manner. In untreated cells, MEG3 associated with the periphery of the nuclear speckles but after treatment with DRB, MEG3 localized to its core (Figure 8C,D).

## 4. Discussion

In this study, we examined the dynamics of the lncRNA MEG3 within the nucleus of living cells. We found that MEG3 is associated with nuclear speckles under regular conditions and that this association increases when transcription and splicing are inhibited. MEG3 is a lncRNA that plays a crucial role in several key biological processes such as p53 stimulation [72]. It was previously shown that MEG3 transcripts could be found in nuclear speckles where the MALAT1 lncRNAs are localized [37]. MEG3 is not expressed in all cell types and is expressed at low levels in many cancerous cells. Since it is not expressed in U2OS cells, we could use this cell line for expressing MS2-tagged MEG3 transcripts that can be followed in living cells. The MEG3-MS2 transcript was retained in the nucleus in accordance with endogenous MEG3 behavior [37]. It partially localized to nuclear speckles similar to the localization of the endogenous MEG3 observed in HepG2 cells. A closer look at the association of MEG3 transcripts with the nuclear speckle marker SRRM2 showed two MEG3 populations, where one appeared to associate with the nuclear speckles while the other was nucleoplasmic. This contrasts with the MALAT1 transcript, which is closely associated with nuclear speckles. The nuclear speckles have subdomains [7] and it appears that the majority of MEG3 transcripts that do associate with the nuclear speckle localize predominantly to the periphery of the nuclear body.

Transcription inhibition causes the nuclear speckles to change structure and become bigger and rounder, since splicing factors are not released from the nuclear speckles into the nucleoplasm under these conditions, where there is no splicing [6]. In contrast to the splicing factors that accumulate in nuclear speckles under transcription inhibition conditions, MALAT1 is released from the nuclear speckles and disperses in the nucleoplasm [33]. This is also true for the paraspeckle associated NEAT1 lncRNA [30]. Since MALAT1 and MEG3 commonly both localize to the nuclear speckles and at least partly overlap, we examined whether MEG3 would react to transcription inhibition in a similar manner. However, we found that under transcription inhibition, MEG3 became highly localized at nuclear speckles. MEG3 therefore differs from MALAT1 and NEAT1 in this respect. Live-cell movies revealed the relatively fast occurrence of this phenomenon. Within 40 min of transcription inhibition, the ring formations of MEG3 around the nuclear speckles became prominent. Intensity analysis of MEG3 association with the nuclear speckle periphery indicated increased MEG3 signal at the periphery as well as an increased overlap with SRRM2 inside the nuclear speckle. Endogenous MEG3 in other cell lines such as HFF-1 and HepG2 behaved similarly to the over-expressed MEG3 in U2OS cells under transcription inhibition. Furthermore, when the cells were treated with the splicing inhibitor PLB, there was increased MEG3 localization to the nuclear speckles. We conclude, therefore, that the relocalization of MEG3 to nuclear speckles occurs due to the reduced activity of splicing factors in the nucleoplasm and not due to the transcription inhibition.

Nuclear speckles are dynamic structures, where the nuclear speckle components shuttle between the nuclear speckle and the nucleoplasm. We wished to understand the nature of the MEG3 nuclear speckle localization and see if the transcripts were anchored to the nuclear speckles or if the association was reversible. Live-cell imaging revealed that by 30 min after DRB was removed from the cells, most of the ring formation around the nuclear speckles dispersed in the nucleoplasm in a pattern similar to untreated cells. This time frame of 30 min correlates with the time required for MALAT1 to re-accumulate in nuclear speckles after release from transcription inhibition [33], suggesting that these two lncRNAs are mutually exclusive with respect to nuclear speckle association. It is possible that MEG3, like other speckle components, is stored in or around the nuclear speckle during transcription inhibition perhaps by binding to a certain nuclear speckle component. Another possibility is that MEG3 functions as a carrier for other nuclear speckle components, bringing it to the nuclear speckle during transcription inhibition. The tertiary structure of MEG3 is important for proper function [72]. To examine which regions of the transcript take part in the association with the nuclear speckles we generated a truncated version and found that exons 4 to 8 were not crucial for MEG3 localization to the nuclear speckles. Rather, exons 1 to 3 continued to surround nuclear speckles upon transcription inhibition. A part of the previously identified NRE responsible for the nuclear retention of MEG3 [45] is located on exon 3. This portion of exon 3 also led to nuclear speckle localization. It appears that neither core proteins SON or SRRM2 nor splicing factor SRSF7 or lncRNA-binding factor hnRNPK are essential for this localization. A poly(A) RNA signal was detected in the core the nuclear speckles; however, the only other known poly(A)+ RNA found in the nuclear speckle is MALAT1 lncRNA that localizes to the periphery of the nuclear speckle [7]. Super-resolution (STED) microscopy demonstrated that not only is there an increased amount of MEG3 at the nuclear speckle periphery after transcription inhibition, but that MEG3 transcripts can integrate into the nuclear speckle core.

Considering that MALAT1, the other known prominent lncRNA at the nuclear speckle, disperses after transcription inhibition, it has not been possible to measure the dynamics of lncRNAs at the nuclear speckle after transcription inhibition. This could be done with MEG3. FRAP measurements showed partial recovery both in treated and untreated cells. However, the recovery was slower for treated cells, indicating a slower exchange of the MEG3 transcripts that associate with the nuclear speckle. To examine this on single MEG3 RNPs, we tracked RNPs at the nuclear speckle in living cells and found that during transcription inhibition conditions, the RNPs remained associated with the nuclear speckle periphery. This suggests that the recovery which was seen after the FRAP is possibly due to new MEG3 transcripts joining the nuclear speckles. Altogether, it appears that MEG3 continuously localizes to nuclear speckles during transcription inhibition. Once MEG3 lncRNA has reached the nuclear speckle, it appears to remain there. The exact mechanism that underlies this phenomenon and the factors responsible for the association are still unknown and await further study.

## 5. Conclusions

Transcription and splicing inhibition cause not only an increased association of splicing factors with nuclear speckles but also the recruitment of MEG3 lncRNA, in contrast to MALAT1 lncRNA that is removed from nuclear speckles under these conditions. This shows that relocalization of nuclear factors to nuclear speckles is regulated both at the protein and RNA levels and suggests that MEG3 and MALAT1 lncRNAs are mutually exclusive with respect to nuclear speckle association.

## Figures and Tables

**Figure 1 cells-11-01942-f001:**
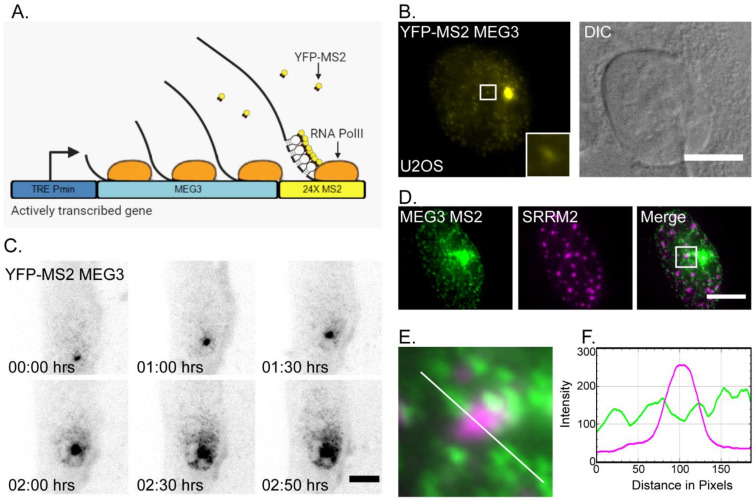
MEG3-MS2 expressed in U2OS cells partially associates with the nuclear speckles. (**A**) Schematic depiction demonstrating the Tet-On MEG3-MS2 expression system. (**B**) U2OS cells stably expressing MEG3-MS2 tagged with YFP-MS2-CP after dox activation (yellow). DIC (grey). The small dots are MEG lncRNPs and the large dot is the active gene. Enlargement of an area with transcripts in the boxed region. Bar = 10 μm. (**C**) Frames from a live-cell movie (Appendix A) of the transcriptional induction (dox) of MEG3-MS2 expression. The images are an inverted version of the transcript tagged with YFP-MS2-CP (black). Images were captured every 10 min for the duration of 3 h. Bar = 5 μm. (**D**) MEG3-MS2 tagged with YFP-MS2-CP (green) is partially associated with nuclear speckles stained with an antibody to the SRRM2 nuclear speckle marker (magenta). (**E**) Enlargement of MEG3 and SRRM2 and (**F**) intensity line analysis from the boxed region in D along the white line. Bar = 10 μm.

**Figure 2 cells-11-01942-f002:**
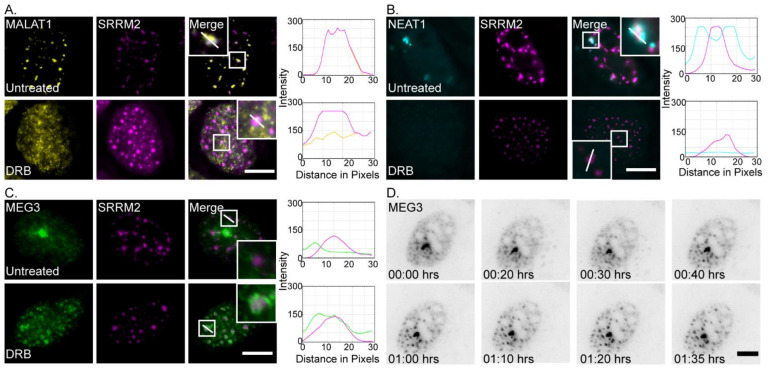
MALAT1 and NEAT1 lose nuclear speckle association in response to transcription inhibition while MEG3 is recruited to the nuclear speckle periphery. (**A**) (Top) RNA FISH to endogenous MALAT1 lncRNA (yellow) overlapped with antibody staining for nuclear speckles marked with SRRM2 (magenta) in untreated cells. (Bottom) When treated with the transcription inhibitor DRB (2 h), MALAT1 lncRNA dispersed in the nucleoplasm and the overlap with SRRM2 was lost. Enlargements of designated areas are in the boxed regions. Right—intensity line analysis from the boxed region along the white line. (**B**) (Top) RNA FISH to endogenous NEAT1 (cyan) overlapped with the periphery of nuclear speckles marked with an SRRM2 (magenta) in untreated cells. (Bottom) In DRB-treated cells, NEAT1 dispersed in the nucleus. Right—intensity line analysis from the boxed region along the white line. (**C**) (Top) MEG3-MS2 tagged with YFP-MS2-CP (green) partially associated with nuclear speckles marked with an antibody to SRRM2 (magenta) in untreated cells. (Bottom) When treated with DRB, MEG3 localized to the nuclear speckles and there was increased association with the SRRM2 signal. Bars = 10 μm. Right—intensity line analysis from the boxed region along the white line. (**D**) Frames from a live-cell movie (Appendix A) of a dox-induced MEG3-MS2 cell expressing the transcript coated with YFP-MS2-CP, starting 15 min after DRB treatment. Images were captured every 5 min for the duration of 1.5 h. The grey image is an inverted version of the transcript tagged with YFP-MS2-CP (black). Bar = 5 μm.

**Figure 3 cells-11-01942-f003:**
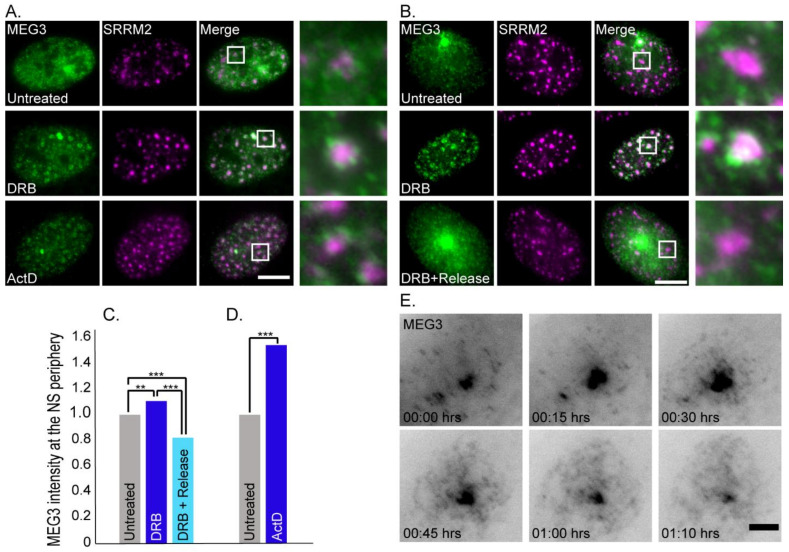
The increase in MEG3-nuclear speckle association in response to transcription inhibition is reversible. (**A**) MEG3 YFP-MS2-CP tagged transcripts (green) and antibody staining of SRRM2 (magenta) under untreated or different transcription inhibition treatments (2 h of DRB or ActD before fixation). Enlargements of the boxed regions appear on the right. (**B**) RNA FISH of MEG3-MS2 transcripts with fluorescent probes to the MS2 region of the lncRNA (green) and antibody staining of SRRM2 (purple). (Top) Untreated cells, (middle) DRB-treated cells (2 h) and (bottom) cells treated with DRB for 2 h and then released in medium for 30 min before fixation. Bars = 10 μm. (**C**) Measurements of the changes in MEG3 intensity at the nuclear speckle (NS) periphery under untreated conditions (#cells = 94, #NS = 1165), DRB treatment (#cells = 61, #NS = 1031), and after 30 min release from DRB (#cells = 108, #NS = 1331) (** *p* < 0.002, *** *p* < 0.0001). (**D**) Measurements of the changes in MEG3 association with nuclear speckles under untreated conditions (#cells = 124, #NS = 1300) and ActD treatment (#cells = 99, #NS = 1024) (*** *p* < 0.0001). Statistical analyses for C and D were performed using a *t*-test. (**E**) Frames from a live-cell movie (Appendix A) of stably expressing MEG3-MS2 YFP-MS2-CP after release from DRB. Images were captured every 5 min for the duration of 1.25 h. The grey image is an inverted version of the transcript tagged with YFP-MS2-CP (black). Bar = 5 μm.

**Figure 4 cells-11-01942-f004:**
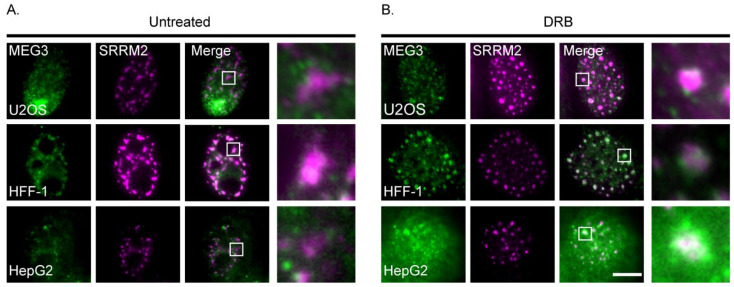
Endogenous MEG3 localizes to nuclear speckles. (**A**) RNA FISH of MEG3 transcripts (green) and antibody staining to SRRM2 (magenta) in (Top) untreated U2OS MEG3-MS2 cells; (Middle) HFF-1s; and (Bottom) HepG2 cells. Enlargements of boxed areas are on the right. (**B**) Same as A with DRB treatment (2 h). Bar = 10 μm.

**Figure 5 cells-11-01942-f005:**
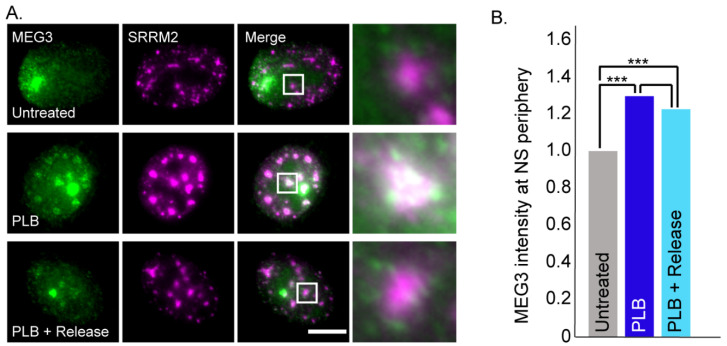
MEG3 surrounds nuclear speckles in response to splicing inhibition. (**A**) RNA FISH of MEG3-MS2 transcripts (green) and antibody staining of SRRM2 in nuclear speckles (purple). (Top) Untreated control cells, (middle) cells treated with PLB for 6 h and (bottom) cells treated with PLB for 6 h and then released 30 min before fixation. Bar = 10 μm. Enlargements of the boxed areas are on the right. (**B**) Plot demonstrating the changes in MEG3 association with nuclear speckles (NS) under untreated conditions (#cells = 84, #NS = 1057), PLB treatment (#cells = 111, #NS = 1212) and after a 30 min release from PLB (#cells = 116, #NS = 1403) (*t*-test, *** *p* < 0.0001).

**Figure 6 cells-11-01942-f006:**
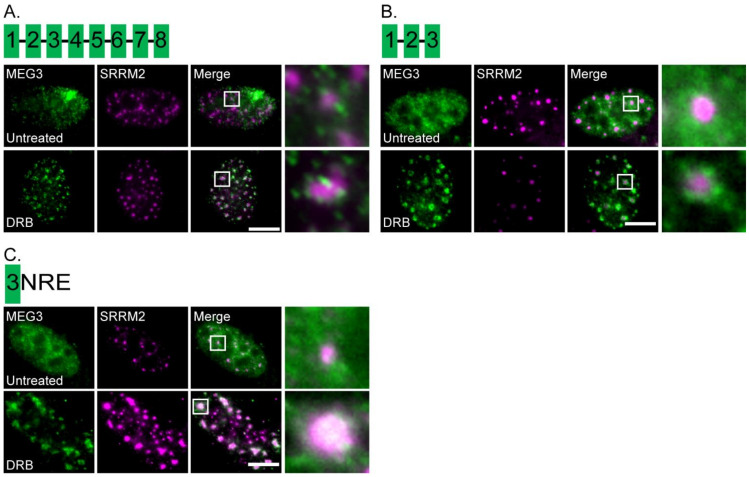
Part of exon 3 is required for MEG3 localization around nuclear speckles during transcription inhibition. (**A**) RNA FISH of full-length MEG3-MS2, (**B**) truncated transcripts (exons 1-3; green), and (**C**) 138 nts of the NRE in exon 3 were expressed in U2OS cells with SRRM2 staining (magenta). The localization of the transcripts was examined in untreated control cells (top) and cells treated with DRB (2 h, bottom). Depictions of the MEG3 exons (green boxes) in each construct are shown above the images. Enlargements of the boxed areas are on the right. Bars = 10 μm.

**Figure 7 cells-11-01942-f007:**
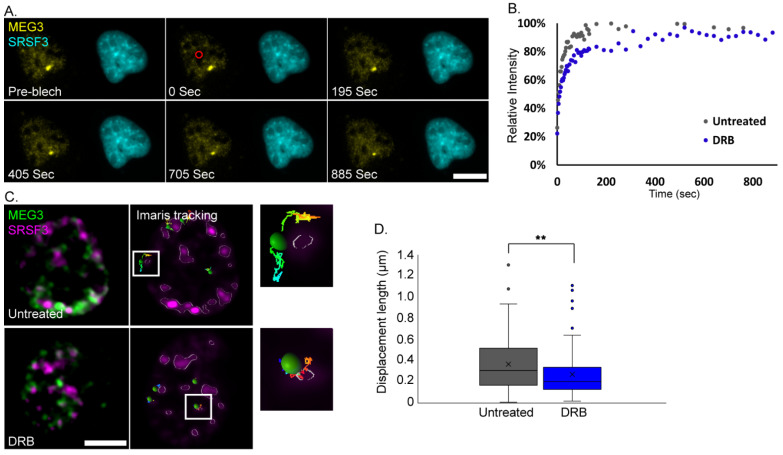
MEG3 is highly associated with nuclear speckles during transcription inhibition. (**A**) Frames from a FRAP experiment showing MEG3 YFP-MS2-CP signal (yellow) and Cer-SRSF3 (cyan) in the same cell. Photobleached area—red circle. (**B**) FRAP recovery curves of MEG3 at nuclear speckles (#untreated cells = 27, #DRB treated cells = 21, * *p* < 0.05 in the middle phase of the recovery curve) in untreated (grey) cells and DRB-treated (blue) cells. DRB-treated cells were imaged during the second hour of the treatment. Statistical analysis was performed using three-parameter asymptotic regression as detailed in the Methods. (**C**) Tracking MEG3 YFP-MS2-CP (green) in relation to the nuclear speckle marker Cer-SRSF3 (magenta). Unprocessed image (left) and Imaris rendering (right) of untreated (top) and DRB treated (bottom) cells. Enlargements of frames from the boxed areas appear in the images. Bars = 10 μm. (**D**) Plot demonstrating the displacement length of MEG3 movements within a 0.25 µm radius of nuclear speckles (NS) under untreated conditions (#cells = 27, #RNA = 108) and DRB treatment (#cells = 25, #RNA = 105) (*t*-test, ** *p* < 0.005). The “x” in the boxplots represents the mean displacement length of each treatment.

**Figure 8 cells-11-01942-f008:**
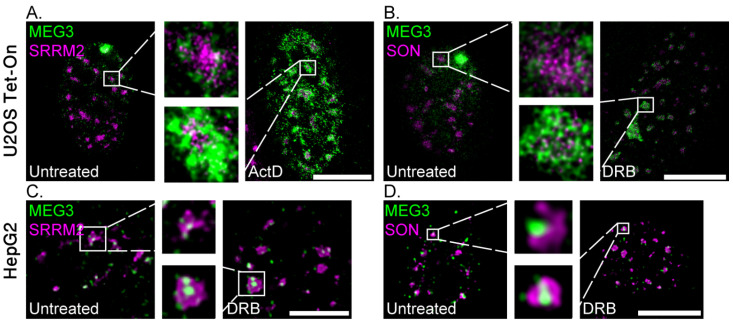
MEG3 can localize to the core of the nuclear speckle under transcription inhibition. (**A**,**B**) STED super-resolution imaging of RNA FISH to MEG3-MS2 RNA (probe to MS2 in green) in U2OS Tet-On cells labeled with (**A**) SRRM2 or (**B**) SON staining (magenta), in untreated cells (left) or with ActD/DRB (2 h) treatments, respectively (right). Enlargements of boxed areas are in the center. (**C**,**D**) Confocal imaging of RNA FISH to MEG3 lcRNA in HepG2 cells labeled with (**C**) SRRM2 or (**D**) SON (magenta), in untreated cells (left) or with DRB treatment (2 h, right). Enlargements of boxed areas are in the center. Bars = 10 μm.

## Data Availability

The datasets generated during and/or analyzed during the current study are available from the corresponding author on reasonable request.

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
