# Peer review of "The Association of MEG3 lncRNA with Nuclear Speckles in Living Cells"

_cells, 2022, doi:10.3390/cells11121942_

Round 1

Reviewer 1 Report

The authors studied the localization and dynamics of the MEG3 to nuclear speckles in a cancer cell line after transcription or splicing inhibition. They generated a fluorescently-tagged MEG transcript to be detected in real-time with high resolution by microscopy. The paper is of great interest and was nicely presented.

I suggest including more data on the Functional significance and linking localization with function. MEG3 demonstrated interaction with PRC2 by RIP and it is well known that MEG3 modulates the activity of TGF-b genes (TGFB2, TGFBR1, and SMAD2) by binding to distal regulatory elements. Thus, it would be interesting to link MEG3 localization with molecular roles in the nucleus.

Discuss the limitations of the study and suggest future research directions.

I suggest including two more figures: i) The biogenesis, structure, and functions of MEG3. ii) a summary figure of the study.

Author Response

The authors studied the localization and dynamics of the MEG3 to nuclear speckles in a cancer cell line after transcription or splicing inhibition. They generated a fluorescently-tagged MEG transcript to be detected in real-time with high resolution by microscopy. The paper is of great interest and was nicely presented.
I suggest including more data on the Functional significance and linking localization with function. MEG3 demonstrated interaction with PRC2 by RIP and it is well known that MEG3 modulates the activity of TGF-b genes (TGFB2, TGFBR1, and SMAD2) by binding to distal regulatory elements. Thus, it would be interesting to link MEG3 localization with molecular roles in the nucleus.
We thank the reviewer for these suggestions. We ordered a series of siRNAs and antibodies to test these options. We depleted EZH2, which is a part of the PRC2 complex, by the use of siRNA, however, no effect was seen on MEG3 nuclear speckle localization (siRNA from IDT and antibody ab228697). In addition, knockdowns using siRNA for nuclear speckle core protein SRRM2 (siRNA from IDT and antibody ab122719), splicing complex component SRm160 (siRNA from IDT and antibody ab221061), and U1 snRNP component SNRPD2 (siRNA from IDT and antibody ab155030) were performed but no effect on MEG3 nuclear speckle localization after DRB treatment was observed.
Discuss the limitations of the study and suggest future research directions.
Not knowing the exact nuclear speckle factor which localizes MEG3 to the structure is the main limitation. This is added to the end of the conclusion section.
I suggest including two more figures: i) The biogenesis, structure, and functions of MEG3. ii) a summary figure of the study.
The journal has asked for a graphical abstract that summarizes the study, so we generated a figure including information along the lines suggested by the reviewer.

Reviewer 2 Report

The current manuscript by Hasenson SE et al. describes the dynamics of MEG3 lncRNA with nuclear speckles using lice cell imaging microscopy techniques. Especially, the tight association of MEG3 with nuclear bodies in contrast to MALAT1 during inhibition of transcription or splicing processes. The results are interesting and critical for understanding the role of MEG3 lncRNA. Overall, the manuscript was well written and appropriate methodologies were used to address the objectives. However, some of the findings were published earlier and authors failed to cite and discuss in the current manuscript. Also, the reviewer has following concerns to be addressed:

Major comments:

  1. Please include a paragraph in second last paragraph of introduction about the rationale for studying MEG3 and its association with nuclear speckles during transcription & splicing inhibition.
  2. Authors have missed one of the important citations from Azam S. et al. (PMID: 31107149), which described MEG3 lncRNA association with nuclear speckles and find out the role of RNP complex proteins and mapped nuclear retention element (NRE) in MEG3 association with nuclear speckles using MEG3 reporter.
  3. In addition to citing the paper, please discuss how the findings from current manuscript were advancing the role of MEG3 lncRNA with respect to Azam et al report.   
  4. Indicate how the cell lines used in the study were procured in section 2.1. Authors just described how the cells were maintained. But, their source and validation (STR profiling and mycoplasma testing) are important for results obtained.
  5. Authors indicated in line-130 about the use of CFP-Clk1 (CDC Like Kinase 1), but the corresponding results were missing in the manuscript. Please correct if it was a typo.
  6. Please make it clear whether the results from Fig.1 (B-C) were just after DOX addition to the media (3h post DOX addition?)
  7. In Figure S1A, please indicate how many replicates used for cell cycle analysis and what statistical test performed. Authors just indicates ‘NS’ and not clear how?
  8. Please indicate how the doses of DRB, ActinomycinD and Amantin were fixed? Cite if any references were followed.
  9. Images in Fig.6 for SRRM2 were quite different from that of other figures with DRB treatment. Fig.6 shows a quite low SRRM2 signals while others indicate a more brighter signals and high punta structures. Please use other cell images where you can show uniform signals like others for SRRM2.
  10. Also add quantification results for Fig.6 like Fig.3 and Fig.5
  11. Authors indicate that last 5 exons were not important for MEG3 nuclear localization (Fig.6). Earlier, a study from Azam et al. identified a NRE element that indispensable for MEG3 nuclear localization (see point 2 above). Does exons 1-3 of MEG3 covers that NRE element?
  12. Also, authors indicated that Exon-3 of MEG3 is long compared to others. Does Exon-3 itself enough for nuclear localization if it has NRE element? Does authors have tested only Exon-3 of MEG3 for localization study?
  13. Authors have tested knocking down SON, SRSF7 and hnRNPK for testing MEG3 localization change upon ActD treatment and are not responsible. But, Azam et al (as indicated in point-2 above), have found out snRNP components are essential for MEG3 reporter localization. If authors could show with knockdown of one or two RNP components using live cell imaging would validate their roe.
  14. Also, authors could test knocking down of nuclear speckle-enriched proteins such as RNPS1, SRm160 and IBP160 for MEG3 localization. These are important for MALAT1 nuclear signals and can be used as control.

Minor comments:

  1. The abbreviations should be expanded at their first usage in the text. Several abbreviations were expanded later than their first usage. For instance MEG3 (in abstract)/ncRNAs/snRNAs (line 43)/FRAP (line 49)/..etc needs to be thoroughly checked and addressed. DRB and PFA were first used in 2.1 section, but expanded in 3.2 and 2.5 sections respectively. Expand ‘4xSSC’ in line-195.
  2. Correct ‘MEG’ in line-89 to ‘MEG3’
  3. Check and correct ‘Dlk-MEG3’ in line-83 to ‘DLK1-MEG3’
  4. Correct ‘hFF (human foreskin)’ to ‘hFF (human foreskin fibroblast) cells’ in line-103.
  5. Authors used hFF (line-103) and HFF (in line-136) for same cell line. Please use uniform abbreviations throughout. HFF-1 is the official name for that cell line.
  6. Correct ‘Son’ to ‘SON’ in line-210
  7. Cite appropriate references for ‘R’ and ‘R-packages’ used in the manuscript in section 2.8.
  8. Adding ‘Abbreviation’ section and some future perspectives in the conclusions would benefit the readers.
  9. Check for accuracy of references cited. For instance ‘Ref.8 DOI has issues’

Best wishes,

Author Response

The current manuscript by Hasenson SE et al. describes the dynamics of MEG3 lncRNA with nuclear speckles using lice cell imaging microscopy techniques. Especially, the tight association of MEG3 with nuclear bodies in contrast to MALAT1 during inhibition of transcription or splicing processes. The results are interesting and critical for understanding the role of MEG3 lncRNA. Overall, the manuscript was well written and appropriate methodologies were used to address the objectives. However, some of the findings were published earlier and authors failed to cite and discuss in the current manuscript. Also, the reviewer has following concerns to be addressed:
Major comments:
1. Please include a paragraph in second last paragraph of introduction about the rationale for studying MEG3 and its association with nuclear speckles during transcription & splicing inhibition.
We thank the reviewer for pointing this out and have added text which explains that since MEG3 is seen at nuclear speckles under regular conditions, and it is known that nuclear speckle protein components change localization during gene expression inhibition, we wanted to examine what would be the fate of MEG3 lncRNA under these conditions. For instance, the lncRNA MALAT1 loses its connection with nuclear speckles under the inhibition conditions and disperses in the nucleoplasm. We were happily surprised to see that MEG3 behaves differently and becomes enriched at the nuclear speckles.
2. Authors have missed one of the important citations from Azam S. et al. (PMID: 31107149), which described MEG3 lncRNA association with nuclear speckles and find out the role of RNP complex proteins and mapped nuclear retention element (NRE) in MEG3 association with nuclear speckles using MEG3 reporter.
We are very embarrassed with this omission. We are of course very familiar with this study on MEG3 by Azam et al and the information within has been useful to us. We think the mention of the work and its citation were erroneously deleted from the text during one of the final edits of the manuscript trying to reduce word count. This has been corrected.
3. In addition to citing the paper, please discuss how the findings from current manuscript were advancing the role of MEG3 lncRNA with respect to Azam et al report.
The study by Azam et al importantly identified and characterized a nuclear retention element that is responsible for retaining MEG3 in the nucleus. In our study, we are able to use transcription inhibition to increase the association of MEG3 with a subregion of the nucleus, the nuclear speckles, and following the reviewer’s suggestion we can now show that the first 138 nts of this element are also connected to nuclear speckle localization in
addition to nuclear retention. Additionally, we move to study MEG3 on a single cell level in living cells.
4. Indicate how the cell lines used in the study were procured in section 2.1. Authors just described how the cells were maintained. But, their source and validation (STR profiling and mycoplasma testing) are important for results obtained.
Available information has been added to the Methods section.
5. Authors indicated in line-130 about the use of CFP-Clk1 (CDC Like Kinase 1), but the corresponding results were missing in the manuscript. Please correct if it was a typo.
This was a mistake and has been removed.
6. Please make it clear whether the results from Fig.1 (B-C) were just after DOX addition to the media (3h post DOX addition?).
Clarification of the dox addition for figure 1B-C has been added.
7. In Figure S1A, please indicate how many replicates used for cell cycle analysis and what statistical test performed. Authors just indicates ‘NS’ and not clear how?
The number of cells used in this experiment has been added to the figure legend of figure S1A.
8. Please indicate how the doses of DRB, ActinomycinD and Amantin were fixed? Cite if any references were followed.
Relevant references have been added.
9. Images in Fig.6 for SRRM2 were quite different from that of other figures with DRB treatment. Fig.6 shows a quite low SRRM2 signals while others indicate a more brighter signals and high punta structures. Please use other cell images where you can show uniform signals like others for SRRM2.
We thank the reviewer for pointing this out. We now present a cell with more uniform SRRM2 signal that represents the observed signal.
10. Also add quantification results for Fig.6 like Fig.3 and Fig.5
We thank the reviewer for this suggestion. Quantification for the short MEG3 containing exon 1-3 was added to fig. S4. There was a statistically significant increase in intensity of MEG3 at nuclear speckles after treatment with DRB compared to control cells in a manner similar to that on the full MEG3 transcript. This would indicate that exon 1-3 is enough for MEG3 nuclear speckle localization.
11. Authors indicate that last 5 exons were not important for MEG3 nuclear localization (Fig.6). Earlier, a study from Azam et al. identified a NRE element that indispensable for MEG3 nuclear localization (see point 2 above). Does exons 1-3 of MEG3 covers that NRE element?
12. Also, authors indicated that Exon-3 of MEG3 is long compared to others. Does Exon-3 itself enough for nuclear localization if it has NRE element? Does authors have tested only Exon-3 of MEG3 for localization study?
These are important points which we failed to describe. Out of the 356 nt long NRE, 138 nts are part of exon 3. It appears from the new data we present that this is enough for MEG3 to localize to nuclear speckles. These findings were further discussed in the section corresponding to figure 6.
13. Authors have tested knocking down SON, SRSF7 and hnRNPK for testing MEG3 localization change upon ActD treatment and are not responsible. But, Azam et al (as indicated in point-2 above), have found out snRNP components are essential for MEG3 reporter localization. If authors could show with knockdown of one or two RNP components using live cell imaging would validate their roe.
We knocked down the U1 snRNP component SNRPD2 which in the article by Azam et al had the highest impact on MEG3 localization percentage wise. However, we did not see any effect on MEG3 localization to the nuclear speckles after treatment with DRB under these conditions. So, we conclude that nuclear retention and nuclear speckle localization are governed by different proteins.
14. Also, authors could test knocking down of nuclear speckle-enriched proteins such as RNPS1, SRm160 and IBP160 for MEG3 localization. These are important for MALAT1 nuclear signals and can be used as control.
We thank the reviewer for these suggestions. We ordered a series of siRNAs and antibodies to test these options. We indeed knocked down SRm160, which is a part of the splicing complex, by the use of siRNA, however, no effect was seen on MEG3 nuclear speckle localization (siRNA from IDT and antibody ab221061). In addition, knockdown using siRNA for nuclear speckle core protein SRRM2 (siRNA from IDT and antibody ab122719), PRC2 component EZH2 (siRNA from IDT and antibody ab228697) and U1 snRNP component SNRPD2 (siRNA from IDT and antibody ab155030) was performed with no effect on the MEG3 nuclear speckle localization after DRB treatment.
Minor comments:
1. The abbreviations should be expanded at their first usage in the text. Several abbreviations were expanded later than their first usage. For instance MEG3 (in abstract)/ncRNAs/snRNAs (line 43)/FRAP (line 49)/..etc needs to be thoroughly checked and addressed. DRB and PFA were first used in 2.1 section, but expanded in 3.2 and 2.5 sections respectively. Expand ‘4xSSC’ in line-195.
Corrected
2. Correct ‘MEG’ in line-89 to ‘MEG3’
Corrected
3. Check and correct ‘Dlk-MEG3’ in line-83 to ‘DLK1-MEG3’
Corrected
4. Correct ‘hFF (human foreskin)’ to ‘hFF (human foreskin fibroblast) cells’ in line-103.
Corrected
5. Authors used hFF (line-103) and HFF (in line-136) for same cell line. Please use uniform abbreviations throughout. HFF-1 is the official name for that cell line.
Corrected
6. Correct ‘Son’ to ‘SON’ in line-210
Corrected
7. Cite appropriate references for ‘R’ and ‘R-packages’ used in the manuscript in section 2.8.
Corrected
8. Adding ‘Abbreviation’ section and some future perspectives in the conclusions would benefit the readers.
Abbreviations have been expanded at their first usage in the text.
9. Check for accuracy of references cited. For instance ‘Ref.8 DOI has issues’
We went over the references and added doi’s to those that have one. The doi of Ref 8 is indeed weird but that is the doi.

Reviewer 3 Report

The authors have examined the dynamics of the MEG3-lncRNA in the nucleus of living cells, using an inducible MEG3 gene that contains elements for tagging RNA. Using this reporter system, the authors have established that MEG3 is a nuclear retained lncRNA. They identified that MEG3 can transiently associate with nuclear speckles under regular culturing conditions and the association with the periphery of nuclear speckles becomes predominant when transcription or splicing are inhibited.

In all, this is a very nice study; the experiments are well-controlled, and the tools produced could be very useful to understand the role of MEG3 in cancer cells and beyond, in non-cancer settings.

A few comments on this study:

  1. MEG3 can translocate in the cytoplasm during depletion of U1 snRNP components SNRPA, SNRNP70 or SNRPD2 (Azam et al. RNA Biol. 2019). The authors should investigate using their reporter system the change in localization of MEG3 in these conditions. In addition, deletion of 356-nucleotide nuclear retention element in the MEG3 truncated transcripts (exons 1-; figure 6) would also help to determine the localization between the nucleus and cytoplasm. This will reinforce the utility of their reporter system to study MEG3 localization.

    Please also comments in the discussion, if there are any other conditions under which MEG3 could localise in the cytoplasm, such as stress or hypoxia.
  2. Which MEG3 transcript has been cloned and what’s the rational for a choice of this?

Minor corrections and editing

Line 45,            Please rephrase to not always start sentence with “Nuclear speckles…”

Line 61, This sentence is ambiguous. how do nuclear speckles then augment the expression of genes?

Line 76, Please correct to say…” revealed function for some of lncRNAs..”

Line 80, Give rational for the choice of lncRNA

Line 89, Please change name MEG to MEG3

Line 134 Please give size of coverslips

Line 164 remove extra “of the”

Line 191 were the probes designed using Stellaris or purchased from them?

Line 211 remove extra space after (Invitrogen )

Line 255  Correct to “there was not”

Line 563           add the word “..possibly due to…” as you haven’t investigated other transcripts

Author Response

The authors have examined the dynamics of the MEG3-lncRNA in the nucleus of living cells, using an inducible MEG3 gene that contains elements for tagging RNA. Using this reporter system, the authors have established that MEG3 is a nuclear retained lncRNA. They identified that MEG3 can transiently associate with nuclear speckles under regular culturing conditions and the association with the periphery of nuclear speckles becomes predominant when transcription or splicing are inhibited.
In all, this is a very nice study; the experiments are well-controlled, and the tools produced could be very useful to understand the role of MEG3 in cancer cells and beyond, in non-cancer settings.
A few comments on this study:
1. MEG3 can translocate in the cytoplasm during depletion of U1 snRNP components SNRPA, SNRNP70 or SNRPD2 (Azam et al. RNA Biol.
2019). The authors should investigate using their reporter system the change in localization of MEG3 in these conditions. In addition, deletion of 356-nucleotide nuclear retention element in the MEG3 truncated transcripts (exons 1-; figure 6) would also help to determine the localization between the nucleus and cytoplasm. This will reinforce the utility of their reporter system to study MEG3 localization.
We thank the reviewer for this suggestion. Knockdown was performed for the U1 snRNP component SNRPD2 which in the article by Azam et al had the highest impact on MEG3 localization percentage wise. However, we did not observe an effect on MEG3 localization to nuclear speckles after treatment with DRB under these conditions. Additionally, out of the 356 nts of the NRE, 138 nts are part of the end of exon 3. Following the reviewer’s suggestion we generated another construct to test the localization issue, which contains the 138 nts of the NRE in exon 3 (although we tried, we did not manage to obtain other clones within the timeframe allowed for the revision) and it appears that this sequence can localize MEG3 to nuclear speckles under DRB treatment.
2. Please also comments in the discussion, if there are any other conditions under which MEG3 could localise in the cytoplasm, such as stress or hypoxia.
The effect on MEG3 under stress conditions is indeed an interesting query. We investigated the effect on MEG3 under oxidative stress by arsenite treatment both alone as well as in combination with ActD. There appeared to be no effect on the MEG3 localization in response to arsenite.
3. Which MEG3 transcript has been cloned and what’s the rational for a choice of this?
We thank reviewer 3 for pointing this out as it was indeed not mentioned. The relevant information on the transcript has now been added. We used isoform 4 since it has a lot of exons in common with the other isoforms.
Minor corrections and editing
Line 45, Please rephrase to not always start sentence with “Nuclear speckles…”
Corrected
Line 61, This sentence is ambiguous. how do nuclear speckles then augment the expression of genes?
Corrected
Line 76, Please correct to say…” revealed function for some of lncRNAs..”
Corrected
Line 80, Give rational for the choice of lncRNA
We thank the reviewer for pointing this out and have added text which explains that since MEG3 is seen at nuclear speckles under regular conditions, and it is known that nuclear speckle protein components change localization during gene expression inhibition, we wanted to examine what would be the fate of MEG3 lncRNA under these conditions. For instance, the lncRNA MALAT1 loses its connection with nuclear speckles under the inhibition conditions. We were happily surprised to see that MEG3 behaves differently and becomes enriched at the nuclear speckles.
Line 89, Please change name MEG to MEG3
Corrected
Line 134 Please give size of coverslips
Corrected
Line 164 remove extra “of the”
Corrected
Line 191 were the probes designed using Stellaris or purchased from them?
The probes were purchased from Stellaris, this was added in the text under the section 2.5.
Line 211 remove extra space after (Invitrogen )
Corrected
Line 255 Correct to “there was not”
Sentence rephrased
Line 563 add the word “..possibly due to…” as you haven’t investigated other transcripts
Corrected

Reviewer 4 Report

The manuscript shows interesting data on long ncRNA localization to nuclear speckles. It could benefit from more discussion of the role of RNA localization and putative mutual exclusive localization of MALAT1 and MEG3 to nuclear speckles.

General comments:

Endogenous MEG3 seems to be expressed in defined areas whereas MEG3 MS2 appears mostly as a large blop in the nuclear periphery under regular conditions. Does this reflect an endogenous property of MEG3 or is it an artefact of the overexpression? Could this affect the results of the experiments?

Could the MS2 tag itself affect response and localization of MEG3 MS2? A control sequence with nuclear localization and MS2 tag would be important for interpreting the results.

In general, the overlap of MEG3 with nuclear speckles is not striking. The association of MEG3 and MALAT1 with nuclear bodies is hard to quantify by eye, and the interpretation would benefit from a quantification of overlap for all figures. Overall the data are hard to interpret.

For ActD this association is quantified and is very small, can this confidently be called biologically significant?

The localization of MEG3 to nuclear speckles seems much better for hFF. Does this reflect an artefact of the MEG3 MS2 construct?

Would the truncation of MEG3 by removing other exons still lead to nuclear speckle localization? The experiments shown support that exons 1-3 can do it, but do not demonstrate that the remaining exons have no function. It would add to the conclusion to have the same experiment without exons 1-3.

Author Response

The manuscript shows interesting data on long ncRNA localization to nuclear speckles. It could benefit from more discussion of the role of RNA localization and putative mutual exclusive localization of MALAT1 and MEG3 to nuclear speckles.

General comments:
Endogenous MEG3 seems to be expressed in defined areas whereas MEG3 MS2 appears mostly as a large blop in the nuclear periphery under regular conditions. Does this reflect an endogenous property of MEG3 or is it an artefact of the overexpression? Could this affect the results of the experiments?
The cells expressing MEG3-MS2 often have large active transcription sites that are easily detectable compared to the transcripts and indeed seem like a 'blob'. This is a known effect of stable transfection of a plasmid that generates a genomic locus with a tandem array of the integrated gene that is highly detectable. This will not be seen when looking at the endogenous transcript, such as MEG3 in HFF and HepG2 cells, because their active genes/alleles are small. Importantly, the free RNA released from the transcription sites has the same pattern in the various cells we used in the study where it is diffuse in the nucleoplasm and somewhat associated with the nuclear speckles. This pattern of MEG3 was also described in Cabili et al 2015.

Could the MS2 tag itself affect response and localization of MEG3 MS2? A control sequence with nuclear localization and MS2 tag would be important for interpreting the results.
We thank the reviewer for suggesting this experiment. We generated a MEG3 construct that does not contain the MS2 repeats, expressed it in untreated and DRB-treated cells, and stained with a probe for the MEG3 sequence. The expression pattern was indistinguishable from the MEG3 transcript with the MS2 repeats in both conditions (fig. S2A). Additionally, figure 4 shows endogenous MEG3 in two other cell lines and the MS2-tagged MEG3 distributes in the same manner.

In general, the overlap of MEG3 with nuclear speckles is not striking. The association of MEG3 and MALAT1 with nuclear bodies is hard to quantify by eye, and the interpretation would benefit from a quantification of overlap for all figures. Overall the data are hard to interpret.
Colocalization analysis has been added to figure 2 in order to make the data easier to comprehend. The graphs show the intensity of the different channels along the white line in the enlargements of the figures. MALAT1 and NEAT1 have significant correlation with the SRRM2 signal in untreated cells whereas MEG3 show a loose correlation adjacent to the SRRM2 signal. After treatment with DRB, MALAT1 and NEAT1 show no correlation with the SRRM2 signal whereas MEG3 showed an increased correlation.

For ActD this association is quantified and is very small, can this confidently be called biologically significant?
Quantification analysis was performed and there is a statistically significant difference in MEG3 localization to nuclear speckles in untreated and ActD treated cells mentioned in fig. 3D.

The localization of MEG3 to nuclear speckles seems much better for hFF. Does this reflect an artefact of the MEG3 MS2 construct?
We believe that the localization is similar for both the endogenous MEG3 and the MS2-tagged MEG3. We speculate that the strong MEG3 signal for HFF can be due to that MEG3 is generally not expressed at high levels in cancerous cell lines.

Would the truncation of MEG3 by removing other exons still lead to nuclear speckle localization? The experiments shown support that exons 1-3 can do it, but do not demonstrate that the remaining exons have no function. It would add to the conclusion to have the same experiment without exons 1-3.
Out of the 356 nts of the NRE, 138 nts are part of the end of exon 3. Following the reviewer’s suggestion we generated another construct to test the localization issue, which contains the 138 nts of the NRE in exon 3 (although we tried, we did not manage to obtain other clones within the timeframe allowed for the revision) and it appears that this sequence can localize MEG3 to nuclear speckles under DRB treatment.

Round 2

Reviewer 1 Report

Thanks to the authors for the effort to add more data with new experiments. I accept that the article will be published.

Author Response

We thank the reviewer's comments.

Reviewer 2 Report

Dear authors,

Thanks for all your efforts to address reviewer concerns in the revised version provided. The new version with added results, quantifications, and statistical analysis sections looks in good shape. I would recommend authors to incorporate the following minor suggestions in the final version of manuscript. 

1) Though authors provided about the source of U2OS cell line, the source of HFF-1 and HepG2 cell lines used in the study is still missing in the section 2.1. Please add those missing details. 

2) Due to the addition of Figure6C results, the corresponding figure legend TITLE needs to be revised. Correct 'last 5 exons' to 'part of exon3 or 138nts in Exon3..)

3) Also, it would be great to revise Figure S4 with quantification data for Figure 6C.

4) Though authors added a general 'statistical analysis' section, it would be better to indicate the kind of test employed in the corresponding figure legends. 

Best wishes,

Author Response

Reviewer 2
1) Though authors provided about the source of U2OS cell line, the source of HFF-1 and HepG2 cell lines used in the study is still missing in the section 2.1. Please add those missing details.

We added the information about the investigators that provided the cell lines.

2) Due to the addition of Figure6C results, the corresponding figure legend TITLE needs to be revised. Correct 'last 5 exons' to 'part of exon3 or 138nts in Exon3..)

Thanks for catching that. We corrected the title.

3) Also, it would be great to revise Figure S4 with quantification data for Figure 6C.

To reach a quantification for 6C would take several weeks to prepare, and since we have only been given several days to revise, we will not be able to provide this in time. Thinking about the data already provided in S4B, we believe these include the data that will arise from a quantification of only part of the exon, and therefore will not likely provide any additional insights. Hence, we hope the reviewer agrees that the conclusions stand as is.

4) Though authors added a general 'statistical analysis' section, it would be better to indicate the kind of test employed in the corresponding figure legends.

Thanks for the suggestion. We added the appropriate information about the statistical analysis used into the figure legends.

Reviewer 4 Report

The authors have addressed the points raised and present an interesting manuscript

Author Response

We thank the reviewer's comments.